# Use of Municipal Solid Waste Incineration Fly Ash in Geopolymer Masonry Mortar Manufacturing

**DOI:** 10.3390/ma15238689

**Published:** 2022-12-06

**Authors:** Ning Lu, Xin Ran, Zhu Pan, Asghar Habibnejad Korayem

**Affiliations:** 1College of Safety Engineering, Chongqing University of Science and Technology, Chongqing 401331, China; 2School of Civil and Transportation Engineering, Hebei University of Technology, Tianjin 300130, China; 3Centre for Infrastructure Engineering, Western Sydney University, Penrith, NSW 2747, Australia; 4School of Civil Engineering, Iran University of Science and Technology, Tehran 1684613114, Iran

**Keywords:** pretreatment, heavy metal leachability, alkali-aggregate reaction, recycling glass, microstructure

## Abstract

The feasibility of partially replacing pulverized fly ash (PFA) with municipal solid waste incineration fly ash (MSWIFA) to produce ambient-cured geopolymers was investigated. The influence of mixture design parameters on the compressive strength of geopolymer paste was studied. The investigated parameters included MSWIFA dosage, the ratio of sodium silicate to sodium hydroxide (SS/SH), the ratio of liquid to solid (L/S) alkaline activator, and the ratio of SH molar. A water immersion method was selected as a pretreatment process for MSWIFA, leading to effectively maintaining the volume stability of the MSWIFA/PFA geopolymer. The mixture of 30% treated MSWIFA and 70% PFA with 12 M SS, 0.5 L/S ratio, and 3.0 SS/SH ratio produced the highest three-day compressive strength (4.9 MPa). Based on the optimal paste mixture, category four masonry mortars (according to JGJT98-2011) were prepared to replace various ratios of natural sand with fine recycling glasses. Up to a 30% replacement ratio, the properties of the mortars complied with the limits established by JGJT98-2011. The twenty-eight-day leaching rate of mortars containing 30% MSWIFA was lower than the limits proposed by GB5085.3-2007. Microstructural analysis indicated that the main reaction product was a combination of calcium silicate hydrate gel and aluminosilicate gel.

## 1. Introduction

Municipal solid waste incineration fly ash (MSWIFA) is recognized as a byproduct generated during the incineration process of municipal solid waste (MSW) in waste-to-energy power plants [1]. The incineration process can be regarded as a sustainable MSW management regime. The generation of MSWI fly ash has rapidly grown and approximately more than 5.8 million tons of MSWIFA was produced in China in 2021. Currently, the most extensively used procedure of MSWIFA disposal is landfilling, which occupies a large area of land. Moreover, MSWIFA contains a variety of soluble salts and heavy metals that are leachable. In extreme conditions, MSWIFA may contaminate soil or groundwater and cause serious environmental threats [2]. Therefore, solidification has been developed as an efficient treatment technology for MSWIFA, and ordinary Portland cement (OPC) is commonly used for the solidification of MSWIFA. The toxic elements in MSWIFA can be effectively immobilized in a binder made with OPC by the actions of physical encapsulation and chemical fixing [1]. For instance, arsenic (As) and lead (Pb), categorized as heavy metals, were successfully immobilized by the formation of Ca_3_(AsO_4_)24H_2_O and Pb_3_(NO_3_)(OH)_5_ through the reaction with hydrated Ca(OH)_2_ existing in cementitious materials [2].

However, using OPC-based solidification technology is associated with high CO_2_ emissions. OPC manufacturing is responsible for a high carbon footprint (0.66–0.82 t CO_2_ per tonne) and high energy consumption [3,4,5]. Geopolymer has been introduced as an alternative low-carbon binder purported to OPC. The benefit of geopolymers is mainly based on their ability to bring high-volume industrial wastes into construction products, leading to a significant reduction in CO_2_ emissions. Geopolymer is manufactured by activation of aluminosilicate byproducts through alkaline additives where the polymerization process can transform them into reaction products [6]. Recently, geopolymer science has been applied to immobilize heavy metals presented in MSWIFA. It has been found that heavy metals were successfully immobilized in geopolymer binders made with various raw materials, including metakaolin [3], red mud [7], coal gangue [8], and pulverized fly ash (PFA) [9]. Among these materials, PFA (a byproduct generated during the burning of coal in power stations) is one of the most commonly used raw materials for manufacturing geopolymers. Zhan and Kirkelund (2021) investigated the solidification of MSWI fly ash by incorporating it into a PFA-based geopolymer binder. The mixture of 80% PFA and 20% MSWIFA activated with 8 M NaOH solution was found optimal under the heat curing condition (24 h at 80 °C temperature). At the age of 28 days, the mixture developed a compressive strength of 15.3 MPa and showed a similar immobilization capacity for heavy metals as compared to the OPC binder. However, the heat-cured regime can be used for the construction of precast members but is regarded as limitated for cast-in-situ applications. Moreover, the previous studies [3,8,9] on MSWIFA/PFA blended geopolymers developed a strength of less than 16 MPa, which does not meet the requirement for structural applications.

Masonry mortar is an efficient bonding material to connect building blocks (including bricks and stones) and seal the gaps between them. The strength required for masonry mortar is not as high as for structural applications. The twenty-eight-day compressive strength of masonry mortar is generally in a range of 5 MPa to 30 MPa. Therefore, masonry mortar manufacturing is likely to constitute an alternative option for recycling MSWI fly ash, although technical and scientific studies on this topic have not been reported in the literature. Masonry mortar is composed of binders and natural sand that have become depleted due to overexploitation as a result of the development of civilization. In the current study, the natural sand was replaced by recycling glass (RG) at different replacement percentages. RG can be categorized as a byproduct generated by crushing mixed-colour bottles and other glass products collected from both municipal and industrial waste streams. Waste glasses are nonbiodegradable and noncombustible. Although the glass recycling rate has reached 50% in China, about 10 million/year of waste glass is still disposed of in landfills. Replacing natural sand with RG could increase the glass recycling rate and reduce the consumption of natural resources in the construction industry.

Overall, it can be concluded that a majority of the previous literature studies on alkali-activated MSWI fly ash have focused on heat-cured geopolymers whereas the engineering properties and matrix formation mechanism (including the reaction products) of ambient-cured MSWI fly ash/PFA-based geopolymer have not been investigated in detail. On the other hand, to our best knowledge, studies on using waste glass in MSWI fly ash/PFA-based geopolymer systems have not been reported in the literature. Thus, the current knowledge of alkali-activated MSWI fly ash is still limited, restricting the use of MSWI fly ash in construction materials.

This work investigates the utilization of waste glass as a partial replacement of natural sand in MSWI fly ash-blended PFA geopolymer binders for the production of masonry mortars. The main aim is to increase the recycling rate of MSWI fly ash, PFA, and waste glass, leading to the promotion of sustainable construction. The specific aims are (i) investigation of effective MSWIFA treatment for reducing volume expansion; (ii) investigation of the effects of mixing parameters on the reaction products and compressive strength (of MSWIFA/PFA geopolymers) via varying the MSWIFA dosages, alkali activators’ compositions, and the ratios of alkali activator to solid; (iii) evaluation of the properties of fresh and hardened MSWIFA/PFA geopolymer mortars with a replacement of up to 30% of natural sand by recycling glass; and (iv) study of the capacity of MSWIFA/PFA geopolymer mortars for immobilization of toxic elements.

The investigation programme was divided into two parts, and the research framework is demonstrated in Figure 1. In the first part, the chemical and physical characteristics of pure MSWIFA, including particle size, morphology, corrosivity, and mineral composites, were characterized. A water immersion method was employed to pretreat MSWI fly ash for solving the volume expansion of MSWIFA/PFA blended geopolymers, which does not require the expensive electrodialytic equipment used in the previous study [9]. Then, the feasibility of preparing the binder with treated MSWIFA and PFA under an ambient curing regime was explored. Next, the effect of the activator nature and MSWIFA content on the compressive strength was analysed. The second part was dedicated to the study of the fresh and hardened characteristics of MSWIFA/PFA geopolymer masonry mortar replacing various dosages of natural sand with recycling glasses. The main concern regarding using glass in cementitious materials is the probable chemical reaction between the silica-rich glass and the alkali present in the pore solution, i.e., alkali–silica reaction (ASR) [10]. The expansion characteristics of MSWIFA/PFA geopolymer mortar bars containing RG were evaluated through the accelerated mortar bar as presented in ASTM C 1260 Standards.

## 2. Experimental Program

### 2.1. Raw Materials

Municipal solid waste incineration fly ash (MSWIFA) was supplied from a waste incineration plant in Chongqing. An ASTM class F fly ash (named PFA) provided by a Henan steam power plant was utilized as another raw material for making geopolymer. The XRD patterns and chemical compositions (detected by X-ray fluorescence, Hitachi High-Tech Analytical Science, Tokyo, Japan) of these two ashes are presented in Figure 2 and Table 1, respectively. It can be seen from Table 1 that CaO and Cl are the main constituents of MSWIFA while Al_2_O_3_ and SiO_2_ are the main constituents of PFA. Figure 2 shows that crystalline phases present in the MSWIFA are CaClOH, NaCl, KCl, and CaCO_3_. The XRD patterns of PFA indicate the reflection peaks of quartz and mullite as well as a small amount of crystalline phase with iron oxides, whose structure is similar to maghemite. As shown in Figure 3, a laser particle size analyser was employed for measuring the particle size distribution of the materials. The average size of MSWIFA and PFA was 57.39 μm and 6.01 μm, respectively.

After cleaning, the collected green beer bottles were ground by a YX-3/100A crusher. The maximum particle size was around 0.3 mm. It is obvious from Figure 2 and Table 1 that the main phase of the recycling glass was amorphous silica with 63.88% SiO_2_ content.

The sand was supplied from Xiamen Aisiou Standard Sand Co., Ltd. (Xiamen, China) Figure 4 illustrates the particle size distribution of the sand, which mets requirements established by the standard of GB178-77 “Standard sand for cement strength test”.

The alkali activator solution used in the experiments was prepared by mixing sodium silicate solution with sodium hydroxide solution. The modulus of sodium silicate solution is 2.2.

### 2.2. Pretreatment of MSWI Fly Ash

When MSWIFA was used as a partial replacement for PFA, the MSWIFA/PFA geopolymer expanded in volume at the early stage of reaction (as shown in Figure 5). The expansion grew with the increment of MSWIFA dosage. The same phenomenon was also observed in the alkali-activated slag system [11]. Previous studies [11,12] have reported that volume instability was often associated with hydrogen produced by the reaction of metallic aluminium according to Equation (1).
(1)$${2\mathrm{Al}+2\mathrm{NaOH}+6H}_{2}O\to {2\mathrm{Na}[\mathrm{Al}(\mathrm{OH})}_{4}{]+3H}_{2}↑\;\mathrm{pH}>11.75$$

According to the above equation, 1/9 g hydrogen gas is produced when 1 g Al is consumed. At ambient temperature, the volume of 1/9 g hydrogen gas is around 3.69 times higher than that of 1 g Al. Moreover, the soluble salts may lead to the deterioration of the durability of cementitious materials containing MSWIFA.

To reduce the soluble salts and metallic aluminium content, commonly used treatment processes include water washing/water immersion treatment [11], acid-base treatment, and chemical reagent treatment [12]. The pH of the MSWIFA leaching solution was as high as 12.06, indicating relatively strong alkalinity. According to Equation (1), a pH value of 12 is sufficient for the initiation of the reaction. Therefore, tape water was used and the immersion treatment method was adopted for removing metallic Al in MSWIFA. The treatment procedures were selected according to Ref. [11]. Water and raw MSWIFA (RMSWIFA) were mixed with a mass ratio of 2:3, and RMSWIFA was fully immersed in water for 72 h. Then, the treated MSWIFA (TMSWIFA) was dried at 105 °C for 48 h.

### 2.3. Sample Preparation

The sample preparation was divided into two stages. The paste samples were prepared in the first stage with an aim to optimize the mix proportions by considering the various mixing parameters. The investigated parameters included MSWIFA dosages, ratios of sodium silicate (SS) solution to sodium hydroxide (SH) solution, concentrations of SH solution, and alkali liquid to solid (L/S) ratio. The MSWIFA dosages were 10%, 20%, 30%, and 40%. The SS/SH ratios were 1.5, 2.0, 2.5, and 3.0. The SH concentrations were 8 M, 10 M, 12 M, and 14 M. The L/S ratios were 0.45, 0.50, 0.55, and 0.60. The second stage was dedicated to preparing mortar samples to evaluate the impact of the inclusion of RG as a sand replacement (0%, 10%, 20%, 30%) on the performance of mortars. The preliminary test results of the current research have shown that the inclusion of RG has little effect on the compressive strength of samples. Therefore, RG is considered a filler rather than a cementitious material in the current study, although 80% of its particle size is less than 75 μm. The proportions of all pastes are summarized in Table 2. The proportions of mortars will be presented in Section 3.5. 

The dry materials (MSWIFA, PFA, RG, and sand) and alkali liquid were mixed in a 40 L Hobart mixer. The dry materials were thoroughly mixed before adding alkali activators. The alkali liquid was then gradually added and mixed continuously for 4 min until a glossy and consistent mixture was reached. A vibration table was used to remove air bubbles during the casting and compaction processes. Before demolding, the samples were kept in the mould for 24 h. The specimens were then cured at ambient temperature under a controlled condition until testing. The paste and mortar were cast into 45 × 45 × 45 mm and 70.7 × 70.7 × 70.7 mm steel moulds to measure compressive strength, respectively. The block of 25 mm × 25 mm × 285 mm was used for assessing ASR potential.

### 2.4. Test Method

(1)Work performance test

Testing of setting in accordance with GB/T1346-2011 “Test methods for water requirement of normal consistency, setting time, and soundness of the Portland cement” was carried out (for paste samples) using the Vicat needle method. The initial and final setting times are reached when the needle is 3~5 mm and 0.5 mm away from the bottom plate, respectively. Mortar water retention and consistency tests were done after mixing, following JCJ/T2009 “Standard for test method of performance on masonry mortar”. The consistency of the mortar was evaluated by the flow table test. All these experiments were performed at a controlled temperature of 23 °C.

(2)Compressive strength test

This test was accomplished on both paste and mortar samples by a hydraulic universal testing machine. For paste samples, the strength was measured at the early age of 3 days while the strength of mortars was measured at 3, 7, and 28 days. JCJ/T2009 standard was employed to conduct this test by adopting a loading rate of 0.25 MPa/s. The average value of three identical specimens was reported as the compressive strength testing result. The calculated standard deviation values were used as error bars length in the corresponding figures. As a percentage, the coefficient of variation (for 90% of results) is less than 10%.

(3)Heavy metal leaching

The mortar samples cured for 28 days were tested for heavy metal leaching concentration according to HJ/T300-2007 “Solid Waste-Extraction procedure for leaching toxicity-Acetic acid buffer solution method”. The crushed particles after strength testing were collected, and the particles were then passed through an 8 mesh sieve. The particles were mixed with deionized water at a ratio of liquid–solid (L/kg) of 20:1. The suspension was then vibrated using a horizontal shaker with the operation at a speed of 30 ± 2 r/min. The shaking was conducted at 23 ± 2 °C for 18 h, and then the leachate was filtered through a 0.45 μm filter. Inductively Coupled Plasma Mass Spectrometer (ICP-MS, Perkin Elmer) was employed to measure the concentration of heavy metal ions in the leaching solution. Whether the leaching concentration exceeded the limitation was judged according to requirements presented in GB16889-2008 “Standard for pollution control on the landfill site of municipal solid waste for hazardous wastes”.

(4)Alkali-silicic reaction expansion

ASR expansion was assessed by the mortar bar method according to ASTM C1260. Just after casting, the moulded geopolymer mortars were cured in a standard curing room for 24 h. Demolding the samples and measuring the length of the samples as the initial length was then carried out. The change in length of mortar bars was measured after alkaline immersion (1 M NaOH solution at 80 °C) at 3, 7, 10, and 14 days. The potential of the aggregate undergoing deterioration (due to ASR) was analysed according to the GB/T 14684-2011 “Sand for construction”.

(5)Spectroscopic/microscopic analysis

After compressive strength test, the crushed specimens were immediately immersed in ethanol to cease the reaction. Before the analysis, the specimens were dried in an oven at 30 °C to achieve a constant weight (moisture content <1%). The spectroscopic analysis was conducted using a high-resolution X-ray diffractometer (Rigaku SmartLab, Tokyo, Japan). The fragments of crushed samples were ground to pass through a 200 mesh sieve. The XRD scan was operated over a range from 10 to 80° 2*θ* with tube setting to 45 KV and 200 mA at a 0.03° step size. In terms of microscopic analysis, a scanning electron with energy-dispersive X-ray spectroscopy (SEM-EDX, HITACHI S-3700N) was employed to investigate the morphology and elemental composition of the geopolymer. Small fragments of crushed samples were coated using gold and fixed on the sample holder with conductive glue.

## 3. Results and Discussion


**
*Part I*
**


### 3.1. MSWIFA Pretreatment

#### 3.1.1. Characteristics of RMSWIFA and TMSWIFA

The heavy metal concentrations in raw and treated MSWI fly ash are tabulated in Table 3. The highest content of heavy metal is Zn followed by Cd, Ba, and Pb. The total heavy element content in MSWI fly ash is influenced by waste source and waste incineration methods. It varies greatly in different countries. According to IAWG (1997) [13], Zn content, Cd content, and Pb content are normally in the range of 9000–70,000 mg/kg, 30–600 mg/kg, and 5300–26,000 mg/kg, respectively. The content of the heavy metals in MSWI fly ash in the current study fell within the above ranges. Among these heavy metals, Cd and Pb content exceeded the limits specified in the standard of heavy metal concentration for construction materials. Therefore, a solidification process is required for safe disposal or using it as a construction material.

Table 4 and Figure 6 present the main components of raw and treated MSWI fly ash and the XRD curves of these samples, respectively. It can be seen from Figure 6 that the intensity of peaks associated with CaClOH disappeared after the soaking treatment. CaClOH is a soluble salt that coverts Ca(OH)_2_ and CaCl_2_ during the soaking process, as shown in Equation (2) [14].
(2)$$2\mathrm{CaClOH}\to {\mathrm{Ca}(\mathrm{OH})}_{2}{+\mathrm{CaCl}}_{2}$$

The XRD pattern of treated MSWI fly ash showed newly detected peaks associated with calcium sulfate hydrate (PDF#39-0725). This is a result of the hydrated reaction of calcium sulfate. The phases, such as chloride, calcium carbonate, and silicon dioxide, in TMSWIFA are roughly the same as those in RMSWIFA with only some minor differences.

As shown in Figure 7, SEM images of raw and treated MSWI fly ash were taken for studying the effects of the soaking treatment on morphology. There is little difference in morphology between the raw MSWI fly ash and the treated MSWI fly ash. The ash particles were loosely packed in the shapes of flocculent agglomerates and flake-like elastics.

#### 3.1.2. Influence of MSWIFA Content on Properties of Paste

A comparison of volume stability between R40 and T40 is shown in Figure 5. Compared with R40, the volume expansion of T40 was significantly reduced. Under 28 days of curing, the volume expansion rates of P100 and T40 were monitored. The volume expansion rate of T40 was around 0.04%, which is comparable to PFA paste without MSWIFA (P100). This suggests that water immersion is an effective method for solving the volume stability of geopolymers containing MSWIFA.

The initial setting time is often used as a quality control measure for the working performance of a mortar. In general, 30 min is the minimum of the initial setting time; otherwise, the fresh mortar would be hard to mould in the required shape in many situations. Figure 8a illustrates the effect of MSWI fly ash content on the initial setting time. In this investigation, the control mix manufactured only by PFA (P100) needed more than 10 h before indicating an early sign of setting. The inclusion of MSWI fly ash significantly accelerated the initial setting time. The higher the content of MSWI fly ash, the shorter the time of the initial setting. This tendency was observed in the samples consisting of both raw and treated MSWI fly ash. When the MSWI fly ash content was the same, mixes with RMSWIFA indicated a shorter setting time than that mixes with TMSWIFA. For example, the initial setting time of the mix containing 30% RMSWIFA was only 5 min while that of the mix containing the same amount of TMSWIFA was 101 min. The results suggest that RMSWIFA as a part of the binder often leads to a rapid setting. When the RMSWIFA dosage is more than 40%, rapid setting occurs, and the mixing procedure cannot be properly completed. On the other hand, the treatment process can effectively prolong the initial setting time of geopolymers containing MSWI fly ash. However, the initial setting time was less than 30 min when the TMSWIFA dosage exceeded 40%. Therefore, the TMSWIFA range used was between 10% and 40%. The initial setting time of all mixes with TMSWIFA was more than 30 min, which meets the minimum initial setting time specified in the relevant standard “General Portland Cement” GB175-2007. Whilst the requirements for the maximum final setting time have been withdrawn from most standards, the final setting time of all mixes (except the control mix) was less than the maximum value of 7 h specified in GB175-2007.

The compressive strength of the blended geopolymers (MSWI fly ash dosage from 10% to 40%) was measured at the age of three days. The results of geopolymers with RMSWIFA and TMSWIFA are presented in Figure 8b. At low dosages (10–20%), the compressive strength of the RMSWIFA mixes reported a higher value than that of the TMSWIFA mixes. At high dosages (30–40%), the compressive strength of mixes containing RMSWIFA was slightly lower than that of mixes containing TMSWIFA. The strength of the blended geopolymers showed no prominent changes when the MSWI fly ash dosage increased from 10% to 40%.

Considering the impact of fly ash content on the compressive strength and initial setting time of the pastes, the meeting of construction requirements, and having a certain strength, the TMSWIFA content considered was 30% in the subsequent tests.

### 3.2. Paste Mixture Design Parameter Selection

After selecting the appropriate dosage of MSWI fly ash, it was necessary to further investigate the influence of mixture design parameters on the properties of the blended geopolymers. According to the existing research, the parameters that affect the performance of geopolymer pastes are the alkali activator liquid–solid ratio, the NaOH concentration, and the ratio of Na_2_SiO_3_ to NaOH solution (SS/SH).

#### 3.2.1. Effect of Liquid–Solid Ratio

The influence of the amount of alkali activator solution was studied in the mixes of T30L45, T30L50, T30L55, and T30L60 with an alkali activator solution of 45%, 50%, 55%, and 60% of total binder, respectively. Figure 9 presents the variation of compressive strength and initial setting time results as a function of alkali activator solution content in the mixes. The higher the amount of alkali activator solution, the longer the setting time of the blended geopolymer with 30% TMSWIFA. As is clear, the initial setting time was delayed by 237.2% when the alkali activator solution content increased from 0.45 to 0.60. Compared to that of the initial setting time, the alkali activator solution content shows less influence on the compressive strength. Increasing the content of the alkaline activator solution by up to 60% resulted in a reduction of only 7.5% in compressive strength.

#### 3.2.2. Effect of NaOH Concentration

The effect of NaOH concentration was investigated in the mixes of T30N8, T30N10, T30N12, and T30N14 containing NaOH concentrations of 8 M, 10 M, 12 M, and 14 M, respectively. Figure 10 illustrates the variation of the compressive strength and initial setting time as a function of NaOH concentration for the alkali activator solution.

The initial setting time decreased as NaOH concentration increased. As illustrated in Figure 10, the initial setting time of the 8 M paste sample was 59 min while that of the 14 M paste sample was 177 min. The compressive strength was also influenced by the NaOH concentration. It can be vividly observed from Figure 9 that raising the NaOH concentration from 8 M to up to 12 M gradually increased the strength of paste samples. However, after growth to 6.5 MPa for the sample containing a NaOH concentration of 12 M, the compressive strength saw a considerable decline to 5.6 MPa with the increment of the concentration from 12 M to 14 M.

#### 3.2.3. Effect of Sodium Silicate to Sodium Hydroxide Ratio

The ratio of sodium silicate to sodium hydroxide was different in the mixes of T30S1.5 (1.5), T30S2.0 (2.0), T30S2.5 (2.5), and T30S3.0 (3.0) to investigate their effects on the compressive strength and initial setting time of geopolymer pastes. For all these mixes, the NaOH concentration and amount of alkali solution were the constant values of 12 M and 50% of the total binder, respectively.

It is apparent from Figure 11 that the higher the content of sodium silicate, the shorter the time of the initial setting. The mix of T30S3.0 had the fastest initial setting time when the sodium silicate to sodium hydroxide ratio by weight was three. On the other hand, the slowest initial setting time of the mix T30S1.5 was observed in the SS/SH ratio of 1.5 (by weight). In this study, among all the investigated parameters, the ratio of SS/SH was the most influential factor affecting the compressive strength of the geopolymer. The compressive strength experienced significant enhancement as the ratio of SS/SH raised from 1.5 to 3.0. It is apparent from Figure 11 that the compressive strength of the S1.5 sample was 2 MPa while that of the S3.0 sample was 4.9 MPa, an increase of 245%.

Considering the effects of mixture design parameters on the early compressive strength and initial setting time of the paste, a liquid–solid ratio of 0.5, NaOH concentration of 12 M, and SS/SH ratio of 3.0 were selected for producing paste specimens for XRD characterization.

### 3.3. XRD Characterization of Geopolymer Pastes

The specimens with and without MSWI fly ash were cured for three days before the test. For specimens with MSWI fly ash, two pastes were prepared by replacing 30% of PFA with RMSWIFA and TMSWIFA, respectively. The XRD diffractograms are reproduced in Figure 12. The crystalline phases, such as quartz (SiO_2_, PDF#086-1560) and mullite (A1_4_SiO_8_, PDF#073-1389), detected in the original PFA still remained in the samples after activation. Compared with the PFA paste, the intensity of peaks associated with mullite declined in the pastes containing RMSWIFA. The peaks related to NaCl were identified in R30 and T30, which is attributed to the large amount of Cl in MSWI fly ash. Compared with untreated raw MSWI fly ash, the CaClOH peak disappeared in T30, indicating that the reaction (according to Equation (2)) may occur under alkali activation.

It should be noted that a diffuse halo between 25° and 40° 2*θ* was observed in all XRD patterns, agreeing with previous studies [15] where this amorphism was commonly linked to an aluminosilicate gel formation (namely N–A–S–H). This suggests that the aluminosilicate gel is one of the major reaction products for both alkali-activated PFA and alkali-activated hybrid binders. In terms of pastes with MSWI fly ash, C–S–H gel formation was identified. MSWI fly ash provides additional calcium, promoting a simultaneous formation of N–A–S–H and C–S–H gels. The previous investigation [16] proved that the coprecipitation of these two gels in a hybrid system is possible. These two gels interact in the process of activation, leading to a dense microstructure and high strength of the binder. Therefore, the strength of the pastes with MSWI fly ash showed a higher value than that of the PFA paste. A comparison of XRD patterns between R30 and T30 showed little difference, explaining why the strength of these two pastes is similar.

### 3.4. Factors Affecting the Initial Setting Time and Compressive Strength of Pastes

Pretreatment is a crucial parameter that influences initial setting time. The initial setting time of the mix with 10% of TMSWIFA was prolonged by approximately 412.8% compared to the mix with the same amount of raw MSWI fly ash. Figure 12 shows that the CaClOH presented in RMSWIFA was removed by the soaking treatment. This could reduce Ca^2+^ concentration in the alkali activation process of TMSWIFA. The Ca^2+^ may react with silicate (from sodium silicate) to accelerate the reaction at an early age [17]. As a result, the initial setting time of the mixes with RMSWIFA was much shorter than that of the mixes with TMSWIFA. Unlike setting time, compressive strength was slightly influenced by the pretreatment. This suggests that free calcium ions may have more influence on the gel formation in the early age of the reaction. In the later age of the reaction, the formation of N–A–S–H gel is chiefly in charge of the material’s compressive strength.

The setting time of the PFA-based geopolymer lasted more than 24 h, but it was not enough to harden for the strength test at three days. The slow rate of geopolymerisation at ambient temperature is responsible for this phenomenon [18] The inclusion of MSWI fly ash led to an acceleration of setting and increase in strength. This could be ascribed to the reaction of extra calcium in MSWI fly ash. The hydration (of calcium ion and silicate) releases heat that could provoke the condensation process of geopolymersiation [19]. Therefore, both the setting and hardening process was promoted as compared to that of the mix without MSWI fly ash. The slight reduction in strength could be seen with a further increment in MSWI fly ash content from 20% to 40%. When the soluble silica content is not enough for maintaining hydration, the excessive Ca^2+^ (in the solution) may have a negative impact on the further dissolution of calcium and reduce the rate of reactions.

Increasing the alkali solution-to-solid ratio led to the presence of excess water in the mix, which slowed the condensation process and eventually increased the initial setting time. Unlike OPC paste where the chemical reaction of water (namely hydration) leads to a strength gain, water in a geopolymer mainly provides the medium for the dissolution and transportation of aluminosilicate speciation [20]. Therefore, the variation of the alkali solution-to-solid ratio from 0.45 to 0.6 only caused a slight decline in compressive strength, which is ascribed to a porous matrix due to the excess water trapped in the hardened mix.

A strong alkali solution is required for the initiation of the dissolution process. In general, a high Na_2_O content is associated with a high reaction degree. However, increasing NaOH concentration does not always increase strength. The strength of the 14 M mix was lower than that of the 12 M mix, as shown in Figure 10. Ref. [21] also found that an activator with a lower molarity led to enhanced strength than a higher molarity. The extremely high pH in the solution may slow the reaction rate by hindering the coagulation and polymerization of the silicate.

Increasing the ratio of sodium hydroxide to sodium silicate accelerated both the setting and hardening process. The high SS/SH ratio increased the soluble silica concentration, which promotes the process of condensation and introduces more Si in the polymeric chain, leading to a denser matrix [22].


**
*Part II*
**


### 3.5. Preparation of Geopolymer Masonry Mortar

Based on the results obtained from pastes, the SS/SH ratio of 3 and molar ratio of 12 were selected for achieving the high strength of mortars. The mix for the maximum strength and optimized setting time was prepared in a binder content of 30% of TMSWIFA, 0.5 L/S ratios, 3 SS/SH ratio, and 12 SH molar ratio. The sand-to-binder ratio was kept constant in all mixes. It should be noted that 0–30% of sand was also replaced by fine waste glass. All mortar mixes proportions are tabulated in Table 5.

### 3.6. Fresh Properties of Masonry Mortar

#### 3.6.1. Workability of Mortars

Workability is commonly described as a mortar behaviour related to the fresh characteristics required during application. In the current study, a consistency test (according to JGJ/T 98-2010) was carried out to evaluate whether or not a mortar mix was workable. The higher the consistency value is, the better flowability of mortars is. According to the provisions of DBJ01-99-2005 “Technical regulations for the application of ready-mixed mortar”, the required consistency value of ready-mixed masonry mortar is between 30 and 120 mm. It can be observed from Figure 13 that the consistency value of all the mortar mixes mets the requirement. The inclusion of fine recycling glass (RG) leds to a decrease in consistency. This agrees with those reported by Zeng et al. with a replacement ratio of 30 using RG [23]. The fine RG had an irregular angular shape (or/and flake-like) and a large aspect ratio, which is prone to friction between particles, leading to a reduction in the consistency of mortars.

#### 3.6.2. Water Retention of Mortars

When the mortar is applied to a substrate, water may be absorbed by the substrate. Reducing free water in the mixture may cause bleeding and segregation, leading to poor bonding between the mortar and masonry units [24]. According to the requirement provided by JGJ/T 98-2010 “Masonry mortar mix proportion design regulations”, Figure 14 shows that the water retention of ready-mixed mortar should not be less than 88%. The incorporation of fine RG had a slight influence on water retention. All mortar mixes showed a water retention value higher than 88%. A mortar with good water retentivity could remain plastic long enough to allow the masonry units to be aligned and plumbed.

### 3.7. Hardened Properties of Masonry Mortar

#### 3.7.1. Compressive Strength

The compressive strength as well as its evolution on different days are presented in Figure 15. The four mortar mixes showed the same trend as a typical OPC mortar prepared according to “Masonry mortar mix proportion design code” JGJT98-2011. The compressive strength increased considerably as the curing age increased up to 28 days. The mortar mixed with natural sand showed a compressive strength of 20 MPa. According to JGJ/T-2010 “Masonry mortar mix proportion design regulations”, masonry mortar is characterized by seven strength grades, including M5, M7.5, M10, M15, M20, M25, and M30. The alkali-activated TMSWIFA/PFA mortar could be used as all strength grades except M25 and M30. The compressive strength did not show any significant change due to the replacement of sand with fine recycling glass of up to 20%. When the replacement ratio reached 30%, the enhancement in compressive strength was 19% when compared to the mix without fine recycling glass. This result agreed with that reported by Zhang [25], which attributed the strength increase to the high hardness of recycling glass.

#### 3.7.2. Alkali–Silica Reaction Expansion

The results of the expansion test followed by the requirements of ASTM C 1260 are presented in Figure 16. After 14 days, the PFA/TMSWIFA specimens containing 10% of fine recycling glass and 20% of fine recycling glass showed average expansions of 0.006 and 0.008%, respectively. When the substitution rate reached 30%, the expansion value increased to 0.028%, which is three times greater than that of the 20% substitution rate. This result is similar to that reported in the literature [26]. Shang [26] reported a 0.03% expansion value of the geopolymer mortar made from 25% of fine recycling glass. Recycling glass contains a large amount of silica that can react with aluminium (from PFA) under alkaline conditions, resulting in the formation of a geopolymer gel. High RG replacement can prolong the induction period of geopolymerisation, which facilitates the formation of zeolites with intercrystalline pores, leading to the expansion of mortar [27]. Another reason for the expansion could be attributed to chemical reactions between the silica in the glass and the hydroxyl ions in the pore water, resulting in the alkali–silica reaction [28,29,30]. TMSWIFA contains a relatively high amount of calcium that may act as an instigator for ASR. It should be noted that none of the investigated specimens exceeded the ASTM threshold (0.1%).

### 3.8. Heavy Metal Leaching

The amount of leaching heavy metals calculated by the ICP-MS inductively coupled plasma mass spectrometer is tabulated in Table 6. The treated MSWI fly ash illustrated a relevant high concentration of Cd and Pb, which are 4.2 mg/L and 0.48 mg/L, respectively. The concentration of these heavy metals exceeded the limitation required by the GB5085.3-2007, raising concern about the use of MSWI fly ash in construction materials. The concentration of heavy metals in mortars with MSWI fly ash is also demonstrated in Table 5. It is apparent from Table 5 that all mortar mixes showed a much lower heavy metal concentration as compared to the raw materials, indicating that the alkali activation could promote the solidification of heavy metals. When increasing the recycling glass substitution ratio, depending on the metal type, the leaching concentration generally decreased except in Zn and Ba. The leaching concentration of these two metals raised when the substitution ratio expanded from 0 to 10%. Then the concentration reduced when the substitution ratio further jumped up to 30%. All mortar mixes indicate that the heavy metal content was far below the limitation, suggesting that treated MSWI fly ash has the potential to be used as raw materials to produce masonry mortar.

### 3.9. Mechanism Analysis

#### 3.9.1. SEM Results

The SEM images of twenty-eight-day cured mortar samples (with various recycling glass content) are illustrated in Figure 17. From Figure 17, the PFA particles have a regular spherical shape that is easily identified. The large unreacted PFA particles were firmly embedded in the amorous phase. Under alkali activation, alumina and silica ions can react with Ca^2+^ to form C–A–S–H gel, which is chiefly responsible for the development of the strength and solidification of heavy metals. A similar morphology can be observed in mortars containing various RG content. In Figure 17b,d, the interface between the glass and the matrix can be vividly observed. It should be noted that the formation of ASR gel was not observed at the glass-matrix interface. This demonstrates that the expansion of FWG 30% could be due to the active geopolymersiation rather than the formation of ASR gel.

#### 3.9.2. XRD Results

Figure 18 presents the XRD patterns of alkali-activated PFA/TMSWIFA mortars with different RG contents. It is apparent that some minerals were found in the mixture regardless of whether the mixture was with or without RG. For example, quartz (SiO_2_, PDF#75-0443) and mullite (Al_5_SiO_10_, PDF# 84-1205) are residual minerals presented in the raw materials (PFA and MSWI fly ash). Compared with the XRD pattern of the paste, peaks of the main phases in the mortar remained the same, indicating that the sand was not actively taking part in the chemical reactions during alkali activation. The NaCl peaks disappeared when the sand was partially replaced by recycling glass. On the other hand, the peak intensity of around 29.2 increased in the mortar mixtures containing glass. This peak was caused by the formation of hydrocalumite, which was often detected in the AAM system [31]. The CaO content of the RG was around 8% (Table 1), which is higher than that of PFA. Therefore, the extra calcium from the glass promoted the formation of hydrocalumite, which is one of the typical layered hydroxides. Lee et al. [32] stated that chloride could be uptook by hydrocalumite as the interlayer species. Apart from the dissolution of the NaCl, the diffraction peaks of phases contained in Mix T30G0 remained relatively unchanged in the mortar mixes with glass, suggesting that the glass mainly acts as a filler. However, strength gain was observed in T30G30 when compared to the mortar without glass. This is in agreement with the previous study, where Zhang et al. [33] also observed an improvement in the strength of the specimen with a high glass replacement. They believed that increasing the glass amount could provide a sufficient ionic concentration in the solution and thus accelerate the formation of initial reaction products. The fast formation of reaction products around unreacted particles led to a long induction period, which allowed for available alkali and other useful species to penetrate through. This will promote the formation of amorphous aluminosilicate gels, leading to an increase in strength.

## 4. Conclusions

This study proves the water immersion method is a promising way to mitigate volume instability, thus promoting an amount of MSWI fly ash to be used in construction materials. In this investigation, the influence of the MSWIFA dosage, SS/SH ratio, L/S ratio, and SH concentration on the compressive strength of MSWIFA/PFA geopolymers were systematically determined. The XRD and SEM analyses were performed to understand the strength development mechanism based on the reaction products and micromorphology. Moreover, this study also evaluates the feasibility of partially replacing natural sand with RG in geopolymers for making masonry mortars. According to the results of this study, the main conclusions are as follows:

This study analyses the feasibility of partially replacing PVA with treated MWSIFA in geopolymers for making masonry mortars. Four mortars were prepared with various sand replacement ratios using recycling glass. According to the results of the study, the main conclusions are as follows:(1)MSWIFA has a high pH of 12.06 which provides sufficient OH^−^ in the water immersion process for removing the metallic Al existing in the ash. The volume expansion of the specimens prepared with the treated MSWIFA was considerably mitigated compared to the specimens prepared with the raw MSWIFA.(2)The initial setting time of mixes with RMSWIFA was much shorter than that of mixes with TMSWIFA. This is due to the fact that Ca^2+^ from CaClOH may react with silicate (from sodium silicate) to accelerate the reaction at an early age. The water immersion method could remove the CaClOH existing in MSWIFA.(3)SEM images of the MSWIFA blended PFA geopolymer mostly illustrated an amorphous geopolymeric gel and calcium-containing hydration product. The calcium-containing hydration product filled the voids within the geopolymeric matrix, resulting in the reasonable strength development of specimens without heat curing.(4)The inclusion of up to 30% fine recycled glass in masonry mortar production did not remarkably affect the mortars’ properties in the fresh and hardened states except for the ASR potential. When the RG content was higher than 20%, the expansion strain increased obviously. The expansion of the mortar with 30% RG could reach 0.028% after 14 days of alkaline immersion. This value was still lower than the limitation proposed by GB/T 14684-2011 standard.(5)The MSWIFA blended PFA geopolymer mortar with an A/B ratio of 0.5, SS/SH ratio of 3, and SH concentration of 12 M reported the highest twenty-eight-day compressive strength (24.3 MPa) at ambient curing conditions.(6)The concentration of leachable heavy metals of MSWIFA blended PFA geopolymer mortar significantly plummeted to less than 1%. For all curing days, including 7 and 28 days, the concentrations of all six metals were within the limitations presented in the relevant standard.

This research illustrates that using multistream wastes in masonry mortar manufacturing could be a viable alternative that would help increase the recycling rate of MSWIFA, PFA, and RG. However, the chemical compositions and physical properties of MSWIFA and PFA vary from place to place, and the ubiquity of mixing parameters still needs further verification and support. In addition, more attention should be focused on long-term performance in future research. From the authors’ point of view, future research interest in MSWIFA/PFA geopolymer mortars should focus on the two aspects: (1) further investigating the mechanical properties, including flexural and bonding strength; (2) further exploring the durability issues, including shrinkage, efflorescence, permeability, and resistance to freezing.

## Figures and Tables

**Figure 1 materials-15-08689-f001:**
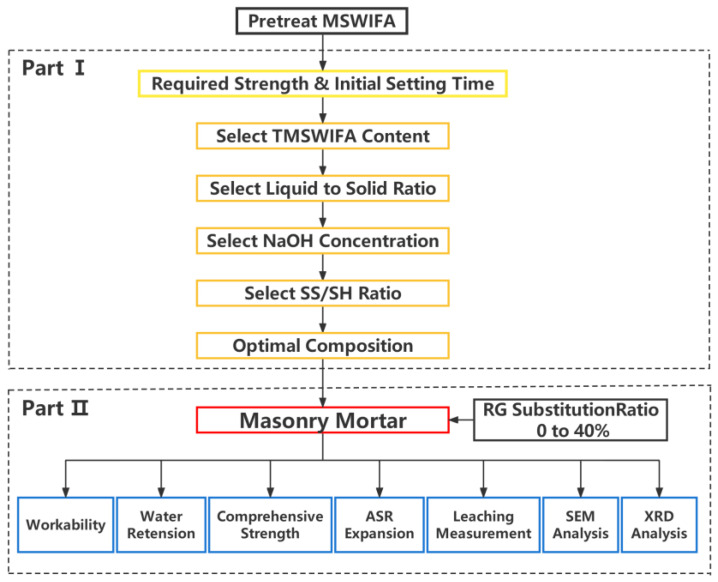
Research framework.

**Figure 2 materials-15-08689-f002:**
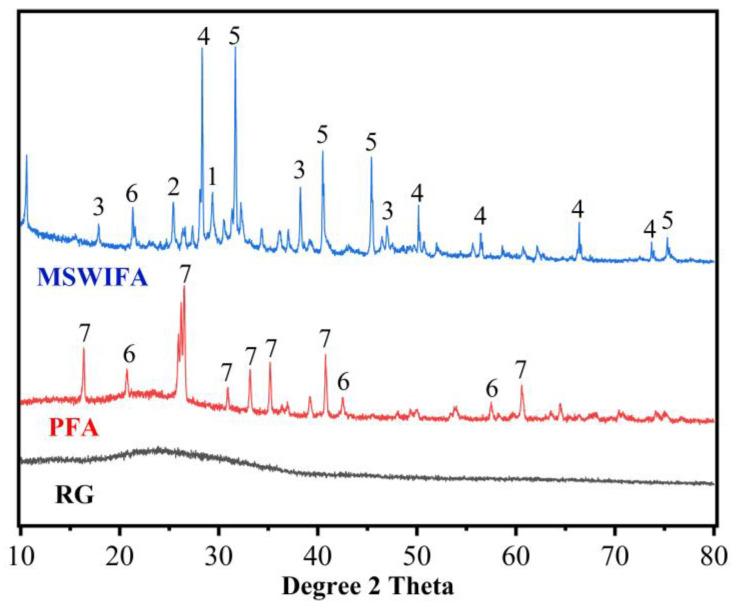
XRD patterns of raw materials (1-CaCO_3_ (PDF#72-1653), 2-CaSO_4_ (PDF#37-0184), 3-CaClOH (PDF#73-1885), 4-KCl (PDF#75-0298), 5-NaCl (PDF#88-2300), 6-SiO_2_ (PDF#79-1906), 7-Mulite (PDF#84-1205)).

**Figure 3 materials-15-08689-f003:**
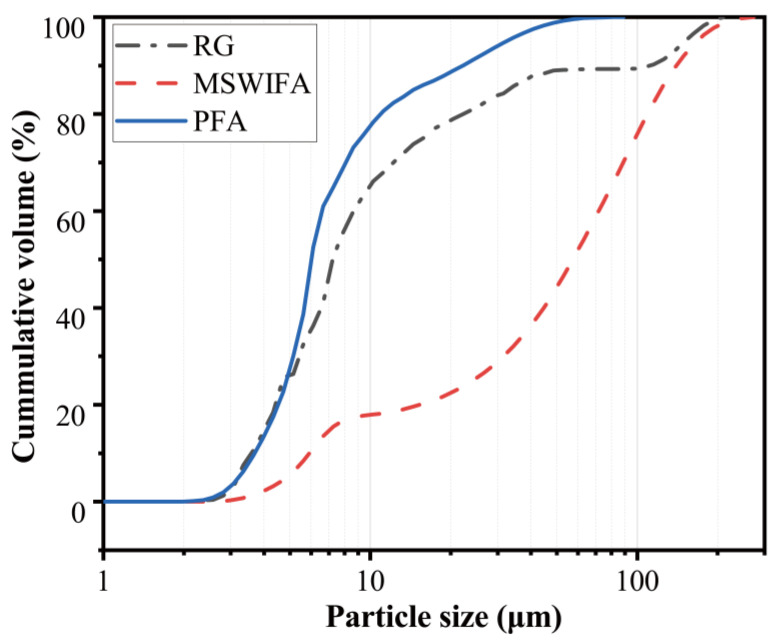
Raw material particle size.

**Figure 4 materials-15-08689-f004:**
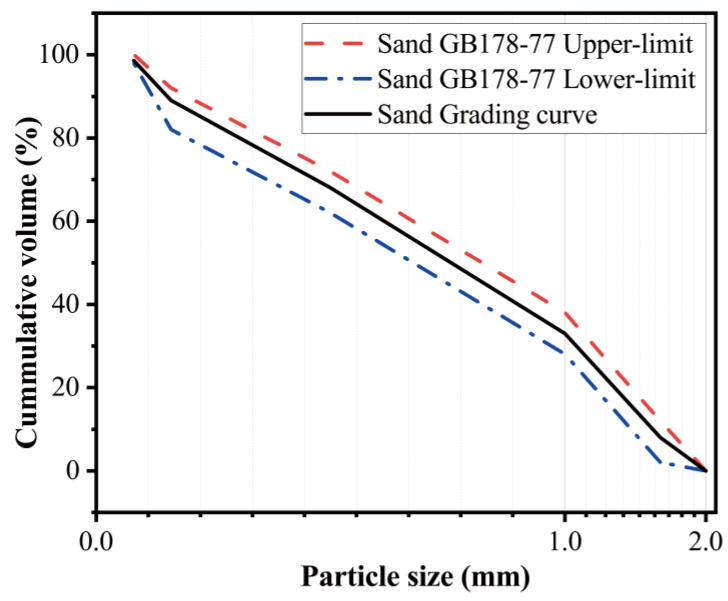
Standard sand particle size.

**Figure 5 materials-15-08689-f005:**
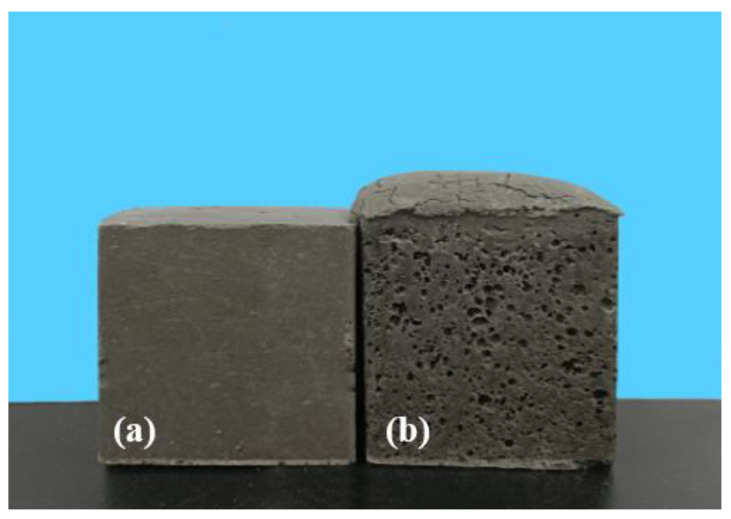
Comparison of cured bodies before (**b**) and after (**a**) fly ash treatment.

**Figure 6 materials-15-08689-f006:**
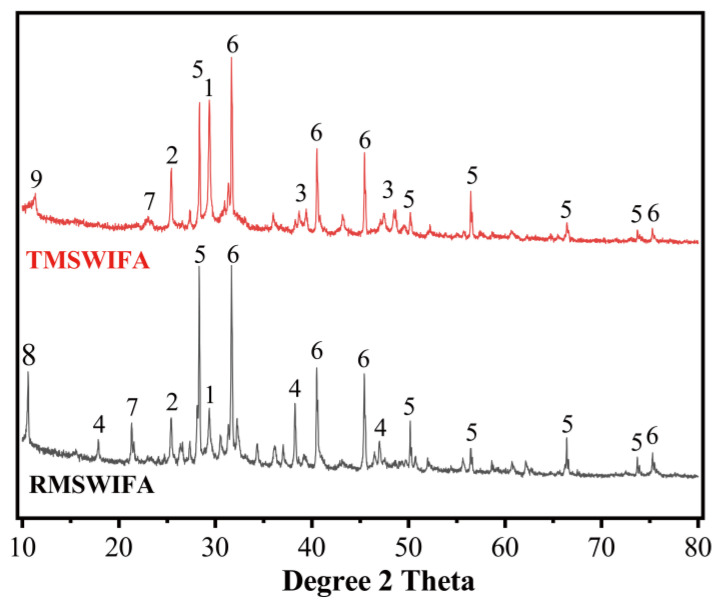
XRD patterns of RMSWIFA and TMSWIFA (1-CaCO_3_(PDF#72-1653), 2-CaSO_4_ (PDF#37-0184), 3-CaSO_4_·H_2_O (PDF#39-0725), 4-CaClOH (PDF#73-1885), 5-KCl (PDF#75-0298), 6-NaCl (PDF#88-2300), 7-SiO_2_ (PDF#79-1906), 8-CaAl_2_Si7O_18_·1.7H_2_O (PDF#21-0132), 9-CaAl_2_Si_7_O_18_·3.5H_2_O (PDF#24-0765)).

**Figure 7 materials-15-08689-f007:**
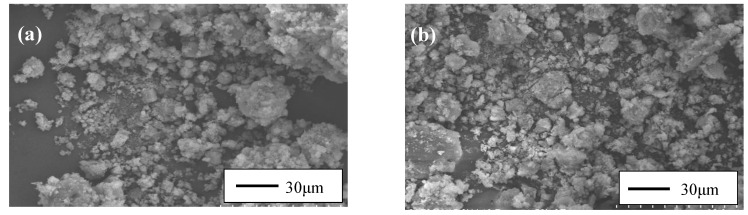
SEM images of RMSWIFA (**a**) and TMSWIFA (**b**).

**Figure 8 materials-15-08689-f008:**
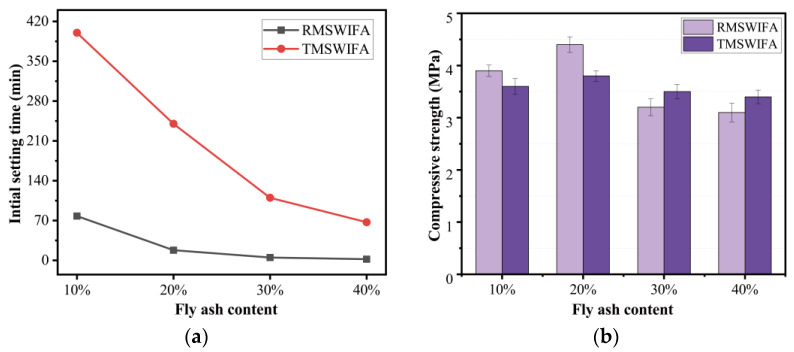
Effect of MSWI fly ash content on paste: (**a**) effect of MSWI fly ash content on the initial setting time of the paste and (**b**) effect of MSWI fly ash content on the paste strength.

**Figure 9 materials-15-08689-f009:**
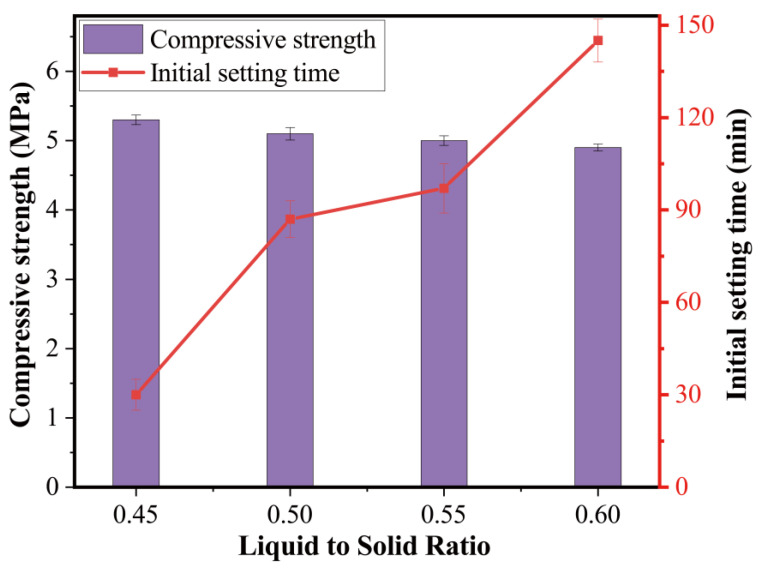
Influence of liquid–solid ratio on the paste.

**Figure 10 materials-15-08689-f010:**
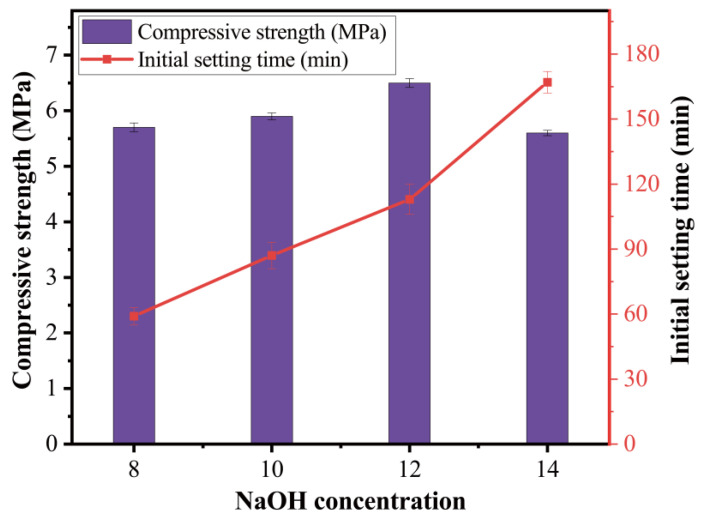
Effect of NaOH concentration on paste.

**Figure 11 materials-15-08689-f011:**
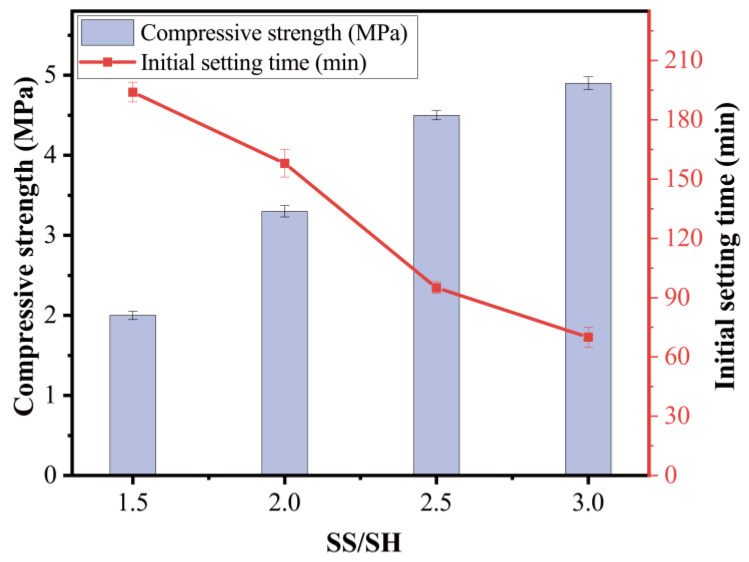
Effect of sodium silicate to sodium hydroxide ratio on the paste.

**Figure 12 materials-15-08689-f012:**
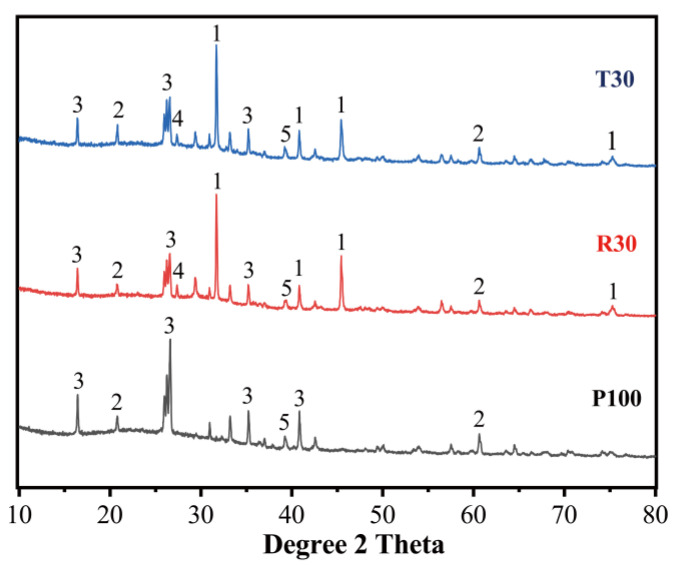
Effect of incorporating MSWIFA on XRD patterns of pastes (1-NaCl (PDF#88-2300), 2-SiO_2_ (PDF#75-0443), 3-Mullite (PDF#84-1205), 4-C-S-H (PDF#14-0035), 5-Na_6_Al_4_Si_4_O_17_ (PDF#39-0299).

**Figure 13 materials-15-08689-f013:**
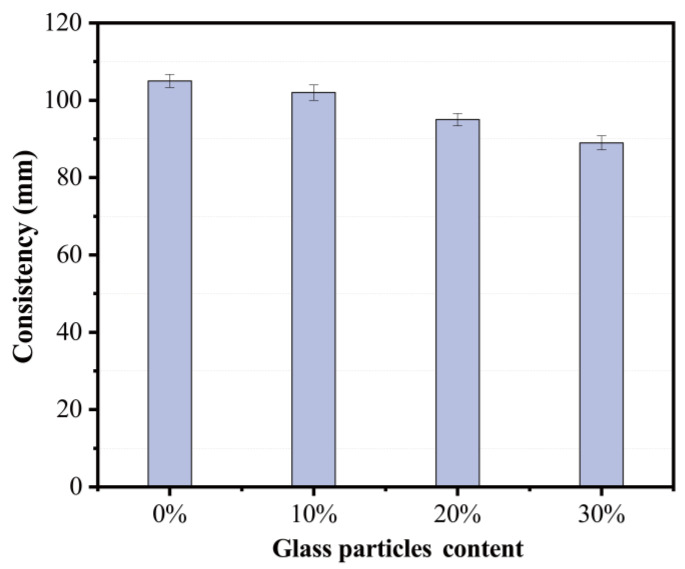
Effect of recycling glass content on mortar consistency.

**Figure 14 materials-15-08689-f014:**
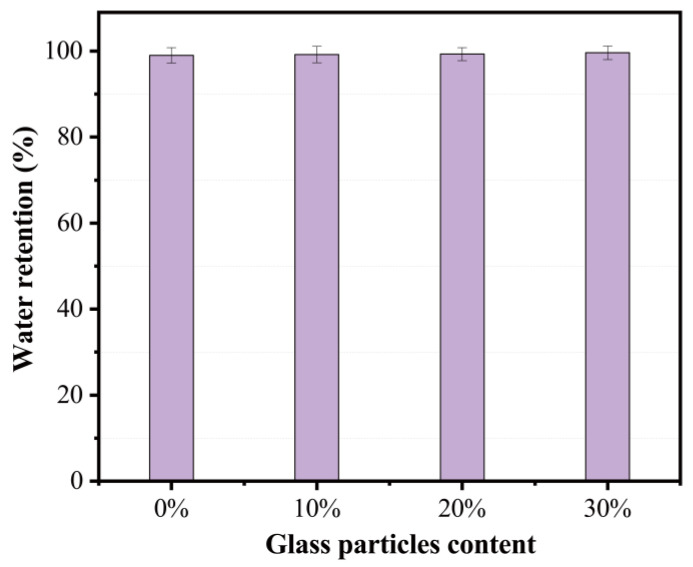
Effect of recycling glass content on water retention of mortar.

**Figure 15 materials-15-08689-f015:**
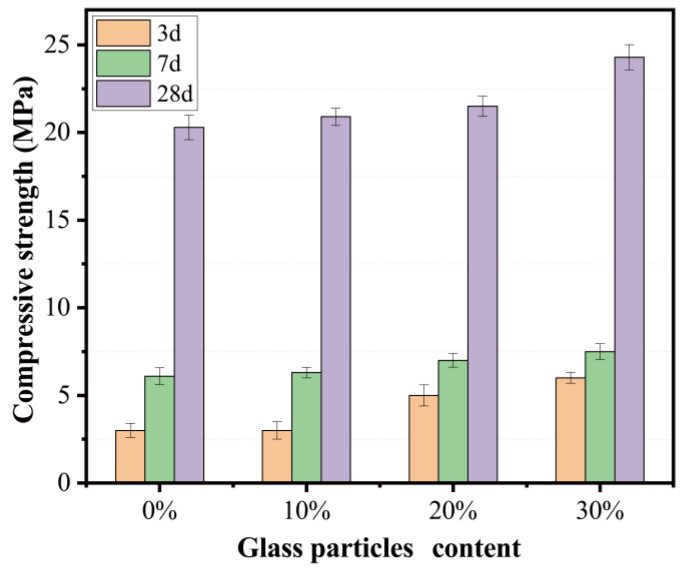
Effect of recycling glass content on mortar strength.

**Figure 16 materials-15-08689-f016:**
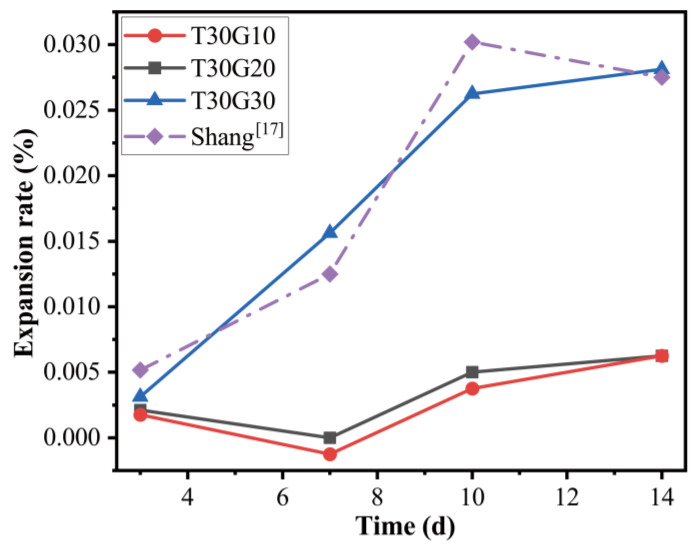
Effect of recycling glass content on the expansion rate of mortar.

**Figure 17 materials-15-08689-f017:**
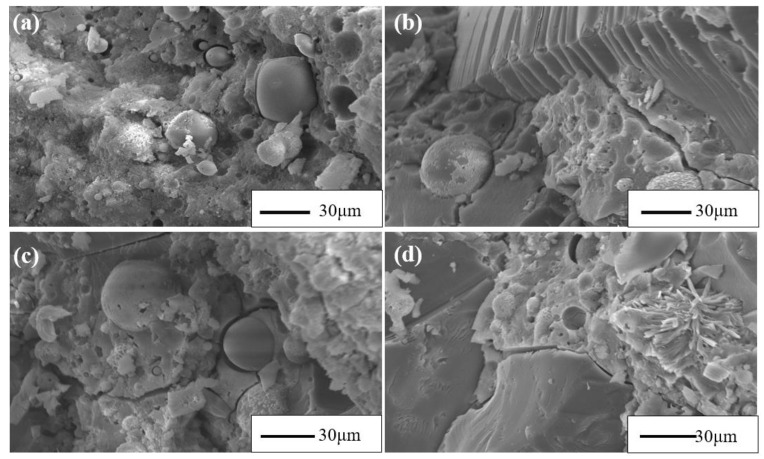
SEM images of different recycling glass contents ((**a**) T30G0, (**b**) T30G10, (**c**) T30G20, (**d**) T30G30).

**Figure 18 materials-15-08689-f018:**
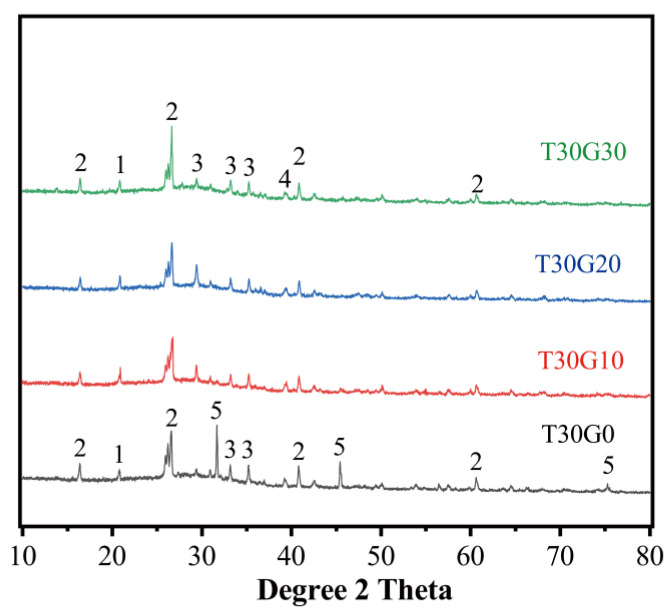
XRD patterns of different recycling glass contents (1-SiO_2_ (PDF#75-0443),2-Mullite (PDF#84-1205), 3-C-S-H (PDF#14-0035), 4- Na_6_Al_4_Si_4_O_17_ (PDF#39-0299), 5-NaCl (PDF#88-2300)).

**Table 1 materials-15-08689-t001:** The chemical compositions of raw materials.

Constituent (%)	SiO_2_	Al_2_O_3_	CaO	MgO	Fe_2_O_3_	Na_2_O	K_2_O	SO_3_	Cl	Loss
MSWIFA	4.95	1.16	43.07	0.91	0.94	10.53	6.37	6.68	23.66	1.73
PFA	45.11	24.21	5.62	0.54	3.35	0.31	2.24	0.69		17.63
RG	63.88	0.45	8.90	0.45	0.63	12.43	0.56			12.52

**Table 2 materials-15-08689-t002:** Mix proposition of paste specimens (by weight).

Mix No.	Designation	RMSWIFA	PFA	TMSWIFA	Liquid-to-Solid Ratio	NaOH Concentration	SS/SH
0	P100	-	100%	-	0.5	10	2.5
1	R10	10%	90%	10%	0.5	10	2.5
2	R20	20%	80%	20%	0.5	10	2.5
3	R30	30%	70%	30%	0.5	10	2.5
4	R40	40%	60%	40%	0.5	10	2.5
5	T10	-	90%	10%	0.5	10	2.5
6	T20	-	80%	20%	0.5	10	2.5
7	T30	-	70%	30%	0.5	10	2.5
8	T40	-	60%	40%	0.5	10	2.5
9	T30L45	-	70%.	30%	0.45	10	2.5
10	T30L50	-	70%	30%	0.50	10	2.5
11	T30L55	-	70%	30%	0.55	10	2.5
12	T30L60	-	70%	30%	0.60	10	2.5
13	T30N8	-	70%	30%	0.5	8	2.5
14	T30N10	-	70%	30%	0.5	10	2.5
15	T30N12	-	70%	30%	0.5	12	2.5
16	T30N14	-	70%	30%	0.5	14	2.5
17	T30S1.5	-	70%	30%	0.5	12	1.5
18	T30S2.0	-	70%	30%	0.5	12	2.0
19	T30S2.5	-	70%	30%	0.5	12	2.5
20	T30S3.0	-	70%	30%	0.5	12	3.0

Label: P = percent of PFA, R = percent of RMSWIFA, T = percent of TMSWIFA, L = Alkali Liquid-to-Solid Ratio, N = NaOH concentration, and S = the ratio of Na_2_SiO_3_ to NaOH solution (SS/SH).

**Table 3 materials-15-08689-t003:** The heavy metal concentrations in raw MSWI fly ash (mg/L).

Heavy Metal	Cd	Pb	Ni	Cu	Zn	Ba	Hg	Se
Raw MSWI fly ash	4.20	0.48	0.02	0.11	76.6	1.06	0.0002	0.004
GB5085.3 leaching limit	0.15	0.25	0.5	40	100	25	0.05	0.1

**Table 4 materials-15-08689-t004:** Composition comparison of raw MSWI fly ash and treated MSWI fly ash.

Constituent (%)	SiO_2_	Al_2_O_3_	CaO	MgO	Fe_2_O_3_	Na_2_O	K_2_O	SO_3_	Cl	Loss
RMSWIFA	4.95	1.16	43.07	0.91	0.94	10.53	6.37	6.68	23.66	1.73
TMSWIFA	5.71	2.88	49.51	1.79	1.59	5.72	4.22	7.12	18.62	2.74

**Table 5 materials-15-08689-t005:** Mortar mix ratio design.

Mix No.	Designation	Cementitious Material		Fine Aggregate		Activator		
		PFA	TMSWIFA	GR	Sand	Liquid-to-Solid Ratio	NaOH Concentration	SS/SH
21	T30G0	70%	30%	0%	100%	0.5	12 M	3.0
22	T30G10	70%	30%	10%	90%	0.5	12 M	3.0
23	T30G20	70%	30%	20%	80%	0.5	12 M	3.0
24	T30G30	70%	30%	30%	70%	0.5	12 M	3.0

Label: T = percent of TMSWIFA, G = percent of RG.

**Table 6 materials-15-08689-t006:** The leaching amount of heavy metals from MSWI fly ash.

Heavy Metal	RG Content			MSWIFA Leaching Limit
	0	10%	30%	
Cd(mg/L)	0.0005	0.0003	0.00028	0.15
Pb(mg/L)	0.0239	0.0168	0.0114	0.25
Ni(mg/L)	<0.0001	<0.0001	<0.0001	0.5
Cu(mg/L)	0.0103	0.00847	0.0075	40
Zn(mg/L)	0.0468	0.0759	0.0265	100
Ba(mg/L)	0.038	0.0716	0.0455	25
Hg(mg/L)	<0.0001	<0.0001	<0.0001	0.05
Se(mg/L)	<0.0001	<0.0001	<0.0001	0.1

## Data Availability

Not applicable.

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
