# Peer review of "Use of Municipal Solid Waste Incineration Fly Ash in Geopolymer Masonry Mortar Manufacturing"

_materials, 2022, doi:10.3390/ma15238689_

Round 1
Reviewer 1 Report
This paper is about the use of MSWIFA in geopolymer masonry mortar. I would like to thank the authors for this practical and very useful article. The work includes a lot of mixes with many variables. It is very well written with minor grammatical mistakes (missing verbs and/or subject in sentences, singular, plural, etc) and a few typos (incorrect/missing punctuation, sentences attached to each other, etc) that need to be taken care of. Kindly find below my comments (the authors forgot to add line numbers to their paper so I will do my best to refer to the locations that need to be corrected)
1- The references mentioned in the text should be mentioned in order. You can’t start with the first ref nb 26 and then jump to ref nb 4. This needs to be corrected throughout the manuscript.
2- Abstract: Line 6 – please replace “ration” with “ratio”.
3- Abstract: please replace “accord to JGJT98-2011” with “according to JGJT98-2011”
4- Introduction – it is well prepared with an extensive literature base.
5- page 2 – please replace “previous study” with “previous studies”.
6- Figure 1 – the vast majority of sentences do not correspond to the figure number in the text. Kindly correct it.
7- Figure 4 – please use the log scale (as in figure 3) for the x-axis.
8- Results and discussion. I really like the results and the discussions that follow.
9- Figure 9 – please insert space in the y axis legend between strength and MPa. Please check all figures throughout the manuscript.
10- section 3.6.2. please rephrase the sentence “.... suggesting that could remain plastic long....”. it is not clear.
11- section 3.7.2 please replace “Alkali-silicic reaction” with “Alkali-silica reaction”
12- section 3.7.2 I would recommend expanding to ref [21] the following articles related to ASR.
doi.org/10.1061/(ASCE)MT.1943-5533.0000086
doi.org/10.1016/j.conbuildmat.2009.12.033
https://doi.org/10.1016/j.cemconres.2022.107007
13- The conclusions correspond to the findings and are well written.
Reviewer 2 Report
The authors have submitted a well prepared paper on the interesting topic of the Use of municipal solid waste incineration fly ash in geopolymer masonry mortar manufacturing. The highlights in this study, the investigated parameters included the municipal solid waste incineration fly ash (MSWIFA) dosage, the ratio of sodium silicate to sodium hydroxide (SS/SH), the ration of the alkaline activator liquid to solid (L/S), and the ratio of SH molar. The paper is clearly presented and provides interesting results. This study is valuable for the practical engineering. However, the following major comments are provided to assist the authors to improve the paper:
1) The article's purpose should be clarified in detail, why this study could be beneficial, and a more in-depth conclusion should be provided.
2) Figure 3 and Figure 4; maintain the aspect ratio of the figure.
3) 2.3 Sample preparation: The brackets for the PFA replacement, ratios of sodium silicate solution to sodium hydroxide solution, concentrations of sodium hydroxide solution, and ratios of alkali liquid to solid are not proper.
4) Why the RMSWIFA range is used between 10% to 40%, Please explain.
5) The sample considered for this study needs to be more convincing. What is the standard deviation and COV value considered?
6) Conclusions: the author should further explain this research's construction application limitations. Please describe in conclusion.
7) Please review the format of the references.
Round 2
Reviewer 2 Report
It has been well revised based on the reviewer' s comments and can be accepted for publication.